# Ethnobotanical Documentation of the Uses of Wild and Cultivated Plants in the Ansanto Valley (Avellino Province, Southern Italy)

**DOI:** 10.3390/plants12213690

**Published:** 2023-10-26

**Authors:** Riccardo Motti, Marco Marotta, Giuliano Bonanomi, Stefania Cozzolino, Anna Di Palma

**Affiliations:** 1Department of Agricultural Sciences, University of Naples Federico II, 80055 Portici, Italy; marco.marotta7@gmail.com (M.M.); bonanomi@unina.it (G.B.); stefania.cozzolino2@unina.it (S.C.); 2Research Institute on Terrestrial Ecosystems, National Research Council (IRET-CNR), 00015 Monterotondo Scalo, Italy; anna.dipalma85@gmail.com

**Keywords:** ethnobotany, traditional plant use, ethnomedicine, medicinal plant, ethnoveterinary, wild food plants, Italy

## Abstract

With approximately 2800 species, the Campania region has the richest vascular flora in southern Italy and the highest number of medicinal species reported in the Italian folk traditions. The study area is inserted in a wide rural landscape, still retaining a high degree of naturalness and is studied for the first time from an ethnobotanical point of view. By analyzing local traditional uses of wild plants in the Ansanto Valley area, the present study aims to contribute to the implementation of ethnobotanical knowledge concerning southern Italy. To gather ethnobotanical knowledge related to the Ansanto Valley, 69 semi-structured interviews were carried out through a snowball sampling approach, starting from locals with experience in traditional plant uses (key informants). A number of 117 plant species (96 genera and 46 families) were documented for traditional use from a total of 928 reports, of which 544 were about medicinal plants. New use reports on the utilization of plants for medicinal (5) and veterinary applications (8) in the Campania region and the whole Italian territory were outlined from our investigations. *Sedum cepaea* is reported as a medicinal plant for the first time in Italy and in the whole Mediterranean basin.

## 1. Introduction

Throughout human history, plants and their derivatives have held significant and purposeful roles in various aspects of human existence. Plants have a significant impact on human nutrition and health and their utilization for medicinal purposes dates back to a time before recorded history, with practices passed down for centuries and in constant evolution [1]. We may consider ethnobotanical knowledge as a part of local ecological knowledge [2]. The traditional ecological knowledge (TEK) of local communities can provide insight into the cultural and ecological importance of different plant species, as well as the potential for sustainable use [3]. In the last years, numerous field ethnobotanical studies have been performed to document the folk uses of plants, with the aim of contributing to the knowledge and conservation of a part of the traditional cultural heritage [4,5]. In particular, Pieroni [6] emphasizes the pressing need for systematic ethnobotanical investigations in southern Europe, particularly in regions that have retained their relative isolation due to historical and geographical factors and where industrial progress has not yet resulted in a complete erosion of their cultural traditions. Currently, traditional indigenous knowledge continues to persist in several regions of the Mediterranean basin, mainly preserved by the elderly living within the surviving rural communities of developed countries [7,8]. Recently, many researchers analyzed the persistence of traditional uses of plants and their products in Italy (e.g., [9,10,11,12]) and particularly in the Mediterranean southern regions (e.g., [13,14,15]). Due to its natural geographic, climatic, and soil conditions, the Campania region exhibits richness in terms of plant diversity and has the richest vascular floras in southern Italy, with approximately 2800 species and subspecies recorded so far [16]. As underlined by Monari et al. [8], Campania has the highest number of medicinal species reported in the Italian folk traditions.

The present study focused on a particular area of the Campania region known as the Ansanto Valley, an impressive place with fascinating geophysical characteristics due to the boiling mud lakes and vents emitting volcanic-like exhalations. Here, the Mefite d’Ansanto arises as the largest non-volcanic natural emission of low-temperature CO_2_-enriched gases ever measured on Earth, with an estimated total gas flux of about 2000 tons per day [17]. At the same time, the site is nestled in a rural landscape, still retaining a high degree of wilderness [18]. In the last few centuries, deforestation has led to several landslides, which, in turn, have led to the discovery of several archaeological remains from a sanctuary sacred to the goddess Mephitis, dating back to the 1st century BC [19]. Recent studies indicate that the potent selection pressure imposed by the extreme environment of the Mefite area can influence population differentiation and adaptation in plants [20]. This study area has never been previously investigated from an ethnobotanical point of view.

Against this background, the present study aims to analyze local traditional uses of wild plants in the Ansanto Valley area, encompassing medicinal, culinary, veterinary, cosmetic, and domestic applications, thereby contributing to the development of ethnobotanical knowledge in Campania and southern Italy. Furthermore, by comparing the available literature data on ethnobotanical studies in southern Italy and in the whole Mediterranean basin, we aim to distinguish new and already described records of the use of a particular plant species in the study area.

## 2. Results and Discussion

We compiled an inventory of 117 taxa (a full list with remarks is given in Appendix A) from 96 genera and 46 families. Based on the interviews, 928 use reports were for all taxa. The use-report number superseded the taxon number because taxa often had more than one use and the same use could be mentioned by more than one informant. They mostly concerned medicinal applications with a percentage of UR equal to 58% of the total UR number, followed by food uses (29%) (Figure 1). These data are in line with other studies on a regional scale (e.g., [14,15]) describing results obtained in similar ethnobotanical research in southern Italy. Asteraceae was the most representative family, counting 18 plant species, followed by Lamiaceae and Rosaceae with 11 and 10 species, respectively (Figure 2a). These families have consistently emerged as the most frequently utilized in folk traditions for medicinal purposes in southern Italy (e.g., [15,21,22]) and in the whole Mediterranean basin (e.g., [23,24,25,26,27]). These families are highly diverse and abundant in Europe, providing a wide range of plant species with various properties and uses. The most cited species are *Malva sylvestris* L. (57 URs), *Matricaria chamomilla* L. (54), and *Cichorium intybus* L. (51) (Figure 1b and Figure 2b).

Over 40% of all the plant species recorded from the interviews fell in multiple application categories. This is likely due to the optimization of natural resources as a consequence of a close connection of the people with the local environment [28]. In particular, use reports described a seamless link between plants used for medicinal and culinary purposes, a frequently observed phenomenon [29] that spans across the globe, contributing to the development of functional food and nutraceuticals.

### 2.1. Medicinal Plants

In total, 544 use reports were recorded for medicinal plants. The plant parts most frequently used for medical purposes were the leaves (42%), followed by the flowers (16%), fruits (15%), stems (7%), and seeds. Leaves are often abundant, easily accessible, and rich in bioactive compounds like essential oils, alkaloids, and flavonoids, which may possess medicinal properties. Many traditional remedies involve the use of the leaves due to their wide availability and ease of use. Even the flowers contain diverse bioactive compounds and are employed in traditional medicine for their potential health-enhancing properties. However, flowers are available in smaller quantities than leaves and may pose a challenge in terms of usability. The aerial parts and the whole plants accounted only for a small percentage (4% each), and the remaining parts accounted for 8% overall. Plant use by oral and topical administration and by inhalation accounted for 59%, 36%, and 5%, respectively. Different preparations and processes of administration of medicinal plants for internal use were mentioned, with raw (32%) or boiled (14%) vegetable parts, decoction (28%), and maceration in oil and alcohol (7%) being the most cited. For topical uses, the direct administration of raw or boiled vegetable parts, decoction, and maceration in oil and alcohol were the most cited. *Malva sylvestris* L. (55 URs), *Matricaria chamomilla* L. (53 URs), and *Ruta graveolens* L. (28 URs) were the most cited plant species for medicinal uses (Figure 3a). Ailments pertaining to the gastrointestinal system recorded the highest number of use reports (149 URs), followed by those related to the respiratory (120) and musculoskeletal (82) systems (Figure 3b).

For the most reported species as medicinal remedies, life form, chorology, and comparative uses in Italian traditional medicine are discussed below. Plants are listed following the alphabetic order of the plants’ scientific names.

Garlic (*Allium sativum* L.) is a species of bulbous flowering plants native to southern and central Asia. Garlic is among the oldest cultivated food plants and is still used also for medical purposes [30]. In the study area, garlic is mainly used as an intestinal disinfectant against intestinal worms (5 URs), to regulate blood pressure (4 URs), for stomach and bellyache treatments (5URs), as an antimicrobial (2 UR), and topically for insect bites (5 URs). In traditional medicine, it is usually used for various diseases, such as hypertension [31,32,33], as vermifuge [34,35], an antimicrobial or antiviral [36,37], and as a treatment for insect bites [38,39]. In the Italian folk phytotherapy, garlic is also widely used for healing wounds and burns [40,41].

Common chicory (*Cichorium intybus* L.) is a perennial herbaceous plant native to temperate Europe, Asia, and Africa, and it has now been introduced worldwide. In the study area, common chicory is mainly used as a liver depurative (8 URs) and diuretic (4 URs). Common chicory is widely used in popular medicine as a depurative of the liver [42,43], intestines, and blood [38,44,45] or to regulate blood pressure in the case of hypertension [46,47]. The aerial parts of C. intybus were also used for their diuretic [35,38,48] and digestive properties [49].

Fennel (*Foeniculum vulgare* Mill.) is a hardy, perennial herb native to the Mediterranean basin and is widely naturalized throughout the world. In the Ansanto valley, fennel fruits are mainly used for bellyache treatments (5 URs) and as a digestive (3 URs). Decoctions of its fruits are widely used in folk medicine as a digestive [50,51,52] and carminative [53,54,55] and against colds and the flu [28,56].

Bay laurel (*Laurus nobilis* L.) is a broadleaf evergreen tree native to southern Europe. In the study area, bay laurel leaves’ decoction is used mainly as a digestive (3 URs) and against the flu (3 URs). Bay laurel leaves’ and fruits’ decoctions are widely used to treat a broad range of diseases, mainly for gastrointestinal [11,43,45] and respiratory ailments [47,48,57], and for dysmenorrhea treatments [50,58,59]. Laurel decoction is also used as a mild sedative [47,53,60].

Common mallow (*Malva sylvestris* L.) is a perennial or annual herb native to western Europe, north Africa, and south Asia and is widespread almost throughout the world. Our respondents reported decoctions of common mallow as a useful treatment for bellyaches (14 URs), the flu, colds, and coughs (11 URs), and against stomach aches or as a digestive (10 URs). In traditional pharmacopeia, common mallow is used to treat various ailments, such as those at the respiratory [41,61], gastrointestinal [33,59,62], and urinary [50,63] systems, and to treat toothaches [57,64] and skin diseases [42,49].

Chamomile (*Matricaria chamomilla* L.) is an annual herbaceous plant native to south-eastern Europe and south-western Asia, and it is now widely distributed throughout the world. Chamomile, in the study area, is reported for bellyaches (URs) and stomach aches (12 URs), as a sedative (7 URs), and for dysmenorrhea treatments (4 URs). Besides its well-known sedative effects [62,65,66], chamomile is reported to be an effective herbal remedy as a spasmolytic and carminative [47,67,68], for dysmenorrhea disorders [29,49,69], respiratory ailments [70,71,72], and against eye inflammations [46,53,71].

Corn poppy (*Papaver rhoeas* L.) is an annual flowering plant native to the temperate areas of Europe, North Africa, and West Asia, and it is naturalized all around the world. Corn poppy in the Ansanto valley is mainly reported as a useful treatment of insomnia for children (11 URs). In folk pharmacopeia, it is mainly used as a sedative and to treat insomnia [15,21,73].

Rue (*Ruta graveolens* L. and the similar *R. chalepensis* L.) are subshrubs native to southern Europe, and it is widely cultivated as an ornamental plant in various parts of the world. Rue, in the study area, is mainly used against the flu, colds, and coughs (5 URs), for stomach ache treatments (5 URs), and topically against toothaches (4 URs). Rue is used for gastrointestinal [6,49,68] and respiratory [14,58] ailments. In the Italian ethnobotanical literature, rue is also widely cited as an anthelmintic [74,75,76], but this use was not reported by the people interviewed in the study area.

Sage (*Salvia officinalis* L.) is an evergreen subshrub that originated in the Mediterranean area and naturalized in many areas around the world. Sage, in the study area, is reported mainly against halitosis (4 URs), sore throats (3 URs), and topically for toothaches and to whiten teeth (3 URs). *S. officinalis* is commonly used as an herbal remedy against the flu and other respiratory diseases [13,28,49] and as a digestive [35,77,78]. Leaves are also used, raw or as a decoction, to wash and whiten teeth and for gingivitis treatments [42,62,67].

The elderberry (*Sambucus nigra* L.) is a deciduous shrub native to Europe, and it was introduced into various parts of the world, including eastern Asia, northern America, New Zealand, and the southern part of Australia. Elderberry is mainly reported by our respondents for flu, cold, and cough treatments (8 URs). *S. nigra* flowers are widely used in folk phytotherapy for the treatment of bronchial diseases [7,45], colds and coughs [10,50,63], as a laxative, or for abdominal pains [50,61,62]. As a topical application, the flowers are used for burn treatments [32,74], wounds [66], and rheumatic pains [49,79,80]. Elderflowers are also widely recognized for their health benefits, which encompass protection against degenerative illnesses such as cardiovascular and inflammatory diseases, cancer, and diabetes [81].

Some use reports recorded in the study area were new for Campania and southern Italy. They are listed below, following the alphabetical order of the species.

*Agrimonia eupatoria* L. is reported to be used for burn treatments, as its local name “evera de lu cuotto” (literally, herb of the burn) suggests. In southern Italy, uses of Agrimonia eupatoria have been documented for wound topical treatments [48], while in the Campania region, *A. eupatoria* was reported only for internal uses as an antispasmodic or anti diarrheal [82,83]. Within the entire Mediterranean region, this species is documented mainly for internal applications, such as for sore throat treatment, as an expectorant, as a hepatic anti-inflammatory, an antispasmodic, or against gastrointestinal diseases [27,84,85,86], and for wound treatments [87] or to treat snake bites [88] for external uses.

*Bellis perennis* L. is used in the study area for healing pimples. In Campania, this plant is reported for its internal use as a febrifuge [14], while in southern Italy, it is used as an external analgesic [59]. In Italy, B. perennis is used as an eyewash [89], for sore treatments [35], or for preventing uterine bleeding during or after labor [8]. In Kosovo, it is used for the treatment of skin infections [90], and in Turkey, it is used as a sedative and an antispasmodic [91] or as a diuretic [92]. In Croatia, it is used for stomach diseases [93].

*Ecballium elaterium* (L.) A.Rich. is used in the study area for sinusitis, cold, conjunctivitis and otitis treatment. In Campania, this species is reported only as a purge for the Cilento area [83]. In southern Italian folk phytotherapy, it is used internally as an emetic or externally against toothaches [89,94,95]. In Spain, it is used for treating skin inflammations [96], and in Libya, it is used for hepatitis treatments [97]. The use of this species for the treatment of sinusitis, as reported from the study area, is well known also in the Croatian islands [4] and Turkey [98,99].

*Parietaria judaica* L. was indicated from our surveys as a hepatomegaly treatment. This medical use is new for Campania and for other southern Italian regions, as this plant is generally reported to be used internally to treat other types of diseases, mainly related to the kidney [32,56,58]. Only in Spain is *P. judaica* reported for the treatment of liver disorders [100,101]. In reports from other European countries, it is used for various ailments but never concerning liver pathologies [2,4,102,103].

*Sedum cepaea* L. resulted to be used for healing wounds and for burn treatments by eight informants. Before our survey, this species was never reported in Italy and in the whole Mediterranean basin as a medicinal plant.

The use of *Borago officinalis* L. as an antirheumatic and *Petasites hybridus* (L.) G. Gaertn., B. Mey. & Scherb for pimple treatments were unknown in all Italian popular phytotherapy, with the sole exception of some alpine areas [49,63], which are about 900 km far from our study area.

The use of *Cannabis sativa* L. fibers mixed with eggs and sugar to make bandages was already reported in Campania but only for the Cava de’ Tirreni area (Salerno province [14]), which is quite distant from the study area in terms of geographical position and vegetational and cultural context.

### 2.2. Food Plants

We recorded 271 use reports describing the use of 43 plant species for culinary practices. *Cichorium intybus* L. was the most cited wild food species (22 URs), followed by *Clinopodium nepeta* (L.) Kuntze subsp. nepeta (21), *Borago officinalis* L., (17), *Foeniculum vulgare* Mill. (17), *Laurus nobilis* L. (15), *Sonchus oleraceus* L. (12), and *Beta vulgaris* subsp. *maritima* L. (12). Twenty-eight plant species were used as vegetables in pizzas and soups or were eaten raw in salads. A typical local dish is the so-called yellow pizza (or “blonde pizza”, “pizza sciatizza”, or “pizza and minestra”) that is based on wild or cultivated vegetables such as *Beta vulgaris* L. subsp. *maritima* (L.) Arcang., *Brassica rapa* L., and *Brassica oleracea* L. Namely, the terms “yellow” or “blond” refer to the use of durum wheat which gives the dough its characteristic golden color. In addition to cultivating commonly used aromatic herbs like basil and parsley, the informants indicated that they often collect wild plants to use as spices. In fact, 18 out of 43 plant species were reported for this culinary use. In some cases, food plants are also eaten for their nutraceutical properties, and many species are also commonly utilized as herbal medicines to alleviate a variety of health conditions.

### 2.3. Cosmetic Plants

We recorded 26 use reports for cosmetic uses in the study area. Skin (6 URs) and hair treatments (19 URs) are the most cited uses. The use of *Salvia officinalis* L. as a hair darkener is new for Campania and the whole of Italy, while the use of *Hedera helix* L. for the same treatment is new for the Campania region but was already reported for other Italian regions (e.g., [52]). The use of *Malva sylvestris* L. leaves boiled with butter as an anti-wrinkler has been previously documented in Italy, but only for the Abruzzo region [7]. The use of olive oil derived from frying river fishes to promote hair re-growth appears to be a rather extravagant practice and has never reported in Italy nor elsewhere before our survey.

### 2.4. Plants for Domestic and Craft Uses

The plant species mentioned by the informants as craft plants to make brooms or baskets, such as *Sorghum bicolor* (L.) Moench, *Panicum miliaceum* L., *Genista tinctoria* L., *Arundo donax* L., or *Salix alba* L., pertain to the traditional material culture of the study area. Although synthetic materials have replaced natural ones, many craft plants are still in use today. Handmade baskets or brooms made with, for example, *Arundo donax* or *Salix alba*, are commonly found in local markets. Although some species were reportedly used to make ink, this practice seems to have disappeared, along with the traditional use of smoking wild plants. New for Italy is the use of *Camellia sinensis* (L.) Kuntze and *Rosa canina* L. to dye fabrics.

### 2.5. Plants for Veterinary Uses

Twenty-eight use reports were recorded in total concerning plants for veterinary uses, as feed, or for the treatment of various ailments. The new records for southern Italy are the following: *Achillea millefolium* L. is used as a decoction, is administered against the intestinal parasites of calves, and is cited in Campania only for topical uses against ovine, cattle, and horse scabs [58]. Borago officinalis L. is used raw in feed as a galactagogue for cows. *Cynodon dactylon* (L.) Pers. is used in horse and rabbit feed to make their coat shinier and to increase the immune system. *Hypericum perforatum* L. is used as a decoction to free cow rumen. *Malva sylvestris* L. is used as a decoction, with bran and cornmeal, to promote the expulsion of the placenta. *Marrubium vulgare* L. is topically used for skin infection treatments. *Rubus ulmifolius* Schott stems as a decoction against the coryza of hens. *Sambucus nigra* L. decoction is topically used against mastitis and coat infections.

### 2.6. Plants for Ritual Uses

Although magical and religious practices are progressively disappearing in the study area, as reported by interviewed people, some ritual uses related to plants are worthy of interest. Olive oil, for example, is part of a ritual practice to defeat “Rizubula” (or “Rizibea”, a facial swelling of unidentified etiology). A hen’s feather is dipped in olive oil and the following litany is recited: “quando Gesù Cristo jeva camminando la trovava una donna pè n’anzi. Donna che vai facendo? Io sono la Rizibula che vò camminando e me ne vado addò la carne umana facia arraggia come a nu cane. Donna questo nun ò fà, cu nu bastone t’aggià bastonà. Non mi bastonà! Nu proverbio te voglio mparà: ruoglio de oliva e penna de gallina, Rizibea vattenni via!”, which can be roughly translated to “when Jesus Christ was walking he found a woman in front of him. Woman what are you doing?. I am the Rizibula and I go walking and I go where I make human flesh angry like a dog. Woman don’t do this, I have to beat you with a stick. Don’t beat me! I want to teach you a proverb: olive oil and chicken feather, Rizibea go away!”. Another ritual example concerns the healing of warts, in which there is no direct application of the plant: crosses are engraved on the nodes of the *Triticum durum* (Desf.) wheat stems and then placed on the ground and watered; as the knots rot, the warts will heal.

### 2.7. The Role of Mefite in the Study Area

The mephitis plays a crucial role for the inhabitants of the study area. The presence of very high concentrations of CO_2_ and sulfur in the air make this place accessible only with many precautions. All metal objects, even if placed many hundreds of meters away from the source of exhalations, are subject to corrosion. Both air and mud from the bottom of the Mefite are traditionally used for therapeutic purposes. Some interviewed people reported, for example, that exhalations are used for sore throats and whooping coughs. The muds are used to treat skin diseases, including burns and wounds, and for making beauty masks. In periods of sulfur shortage, treatments against the fungal diseases of vines were made by applying, on the leaves, the water gathered in the Mefite. Traditionally, shepherds from the area or neighboring regions brought their sheep and dogs to the Mefite to treat them with Mefite waters or muds against mange, scabies, rabies, and ticks. From the milk of sheep grazing grass near the Mefite area, and, for this reason, since it contains high concentrations of sulfur products, a cheese is produced locally known as “Carmasciano”, recognized for its peculiar characteristics as a PAT (Traditional Agri-food Product) by the Italian Ministry of Agricultural, Food, and Forestry Policies [104].

## 3. Materials and Methods

### Study Area

The survey area is the Ansanto Valley, located in a territory of the Avellino province known as Irpinia (40°58′ N, 15°08′ E, ~720 m a.s.l.; Campania region, southern Italy), (Figure 4).

The Ansanto Valley is a large intermontane territory, close to the hilly mountainous chain of the Apennines, whose borders, include the municipal territories of Rocca San Felice, Gesualdo, Sturno, Frigento, Villamaina, and Torella dei Lombardi. The territorial context of the Ansanto Valley is still characterized by a rural economy, which preserves a millenary landscape fabric. Within this area there are a few Mefite pools of geothermal manifestations, consisting of boiling mud and cracks in the ground from which lethal gaseous concentrations of CO_2_ arise, along with sulfuric emissions which enrich the area with sulfur, resulting in a distinctive smell of hydrogen sulfide in the air [17]. The landscape at Mefite is marked by striking physical features, with vibrant mineral deposits creating colorful terraces around the geothermal features (Figure 5). The land use is mainly characterized by oak forests, meadows, olive groves, and cultivated fields.

## 4. Ethnobotanical Methodology

Fieldwork was conducted form June 2021 to October 2022 in the municipality of Rocca San Felice and in the surrounding villages of Villamaina, Gesualdo, and Frigento. For the interviews, we selected local experts (key informants) who still retain traditional knowledge of wild plant uses due to their family traditions, occupations, age, or personal interests. Key informants were therefore individuals who could provide valuable information, insights, or perspectives related to the research objectives and who have connections or relationships with the target population under study. Using a snowball sampling approach [105], we requested the informants to indicate further individuals that were likely to provide valuable insights or information related to the research focus and were in representation across different categories such as age, gender, socioeconomic status, or geographic location. Semi-structured interviews were carried out to acquire information on plants (common or scientific names) and their use for medicinal, food, veterinary, cosmetic, and craft purposes, supplemented by informal walks in the field. To prevent any possibility of missing information, two interviewers (one woman and one man) collected data separately. The first interviewer conducted the semi-structured interview, and the second interviewer was responsible for data collection only. Prior to conducting each interview, following the guidelines of best practices in ethnopharmacological research [106,107], the research objectives were explained to the informant and verbal consent was obtained. In total, 69 informants were interviewed, namely 37 women and 32 men, whose age ranged from 26 to 97 years (68 on average).

The identification of the plants indicated by the informants was conducted in the field or in a laboratory, based on dichotomous keys and morphological characteristics reported in Pignatti [108], by using a stereomicroscope and light microscopy if needed. Plant nomenclature followed the World Flora Online [109], and angiosperm families were organized according to the APG IV classification [110]. Abbreviations of authors were standardized as indicated in Brummitt and Powell [111], as recommended by Rivera et al. [112]. Herbarium specimens were deposited in the herbarium of Portici (Department of Agricultural Sciences, University of Naples Federico II) with codes from AV-001 to AV-104, alphabetically ordered.

As indicated by Heinrich et al. [106], and by Weckerle et al. [113], we only presented primary data in our results with the total number of use reports (URs) which represent the number of individual citations of a plant taxon. We set up a database including the taxon (with family), local name, parts used, preparation, administration, use recorded, and the total number of URs. To categorize the diseases treated with plants indicated from the interviews, we adopted a symptom-based nosological approach commonly employed in ethnobotanical research (e.g., [114,115,116]).

A comparative analysis was carried out on the medicinal uses of plants on the basis of the available literature dealing with popular phytotherapy in Italy. Only URs in a number of three or more per plant species were taken into consideration. To compare our results with those reported in the Italian literature, Web of Science, Scopus, and Google Scholar were used as databases, using “Italy”, “ethnobotany”, “ethnobotanical”, “ethnopharmacology”, and “medicinal plants” as key words and “OR” as the connector. Additional papers were selected from references cited in the collected papers. The criteria for article selection were defined a priori to avoid personal biases. We searched both national and international journals published from 1967 to 2021. Publications were filtered for the English and Italian languages, duplicates, document type (no patents), and full-text availability. Books and papers not subject to peer review were not taken into consideration.

## 5. Conclusions

This study provides a comprehensive overview of ethnobotanical knowledge in the Ansanto Valley, revealing the diverse utilization of plant species for medicinal, food, cosmetic, domestic, craft, veterinary, and ritual purposes. The traditional uses of plants reflect the deep-rooted cultural heritage and the significant role of plants in the lives of the local communities. The documentation of novel ethnobotanical findings further enriches our understanding of traditional plant uses in this region, emphasizing the importance of preserving and promoting this valuable knowledge for future generations. The findings of this study can serve as a basis for further research and the development of policies and programs for the conservation and sustainable use of plant resources in the study area and beyond.

## Figures and Tables

**Figure 1 plants-12-03690-f001:**
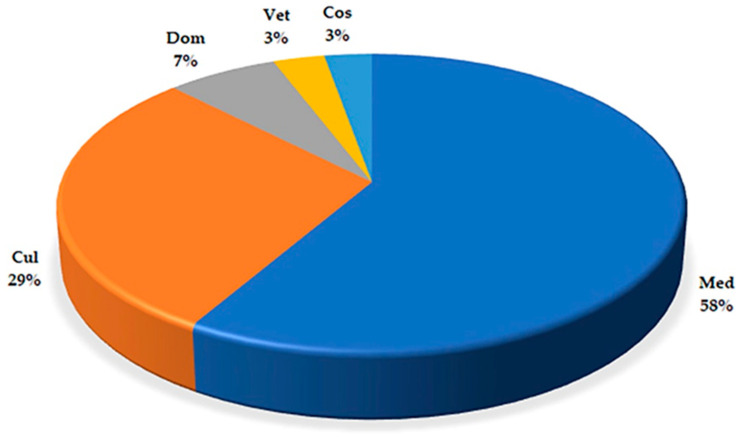
Percent use categories (*n* = 927) among the 116 taxa recorded in the study area (Med: medicinal uses; Cul: food uses; Dom: craft and domestic uses; Cos: cosmetic uses; Vet: veterinary uses).

**Figure 2 plants-12-03690-f002:**
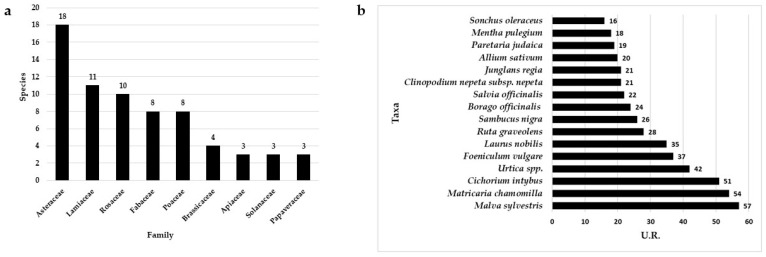
Number of species per family (**a**) and number of use reports per species (**b**).

**Figure 3 plants-12-03690-f003:**
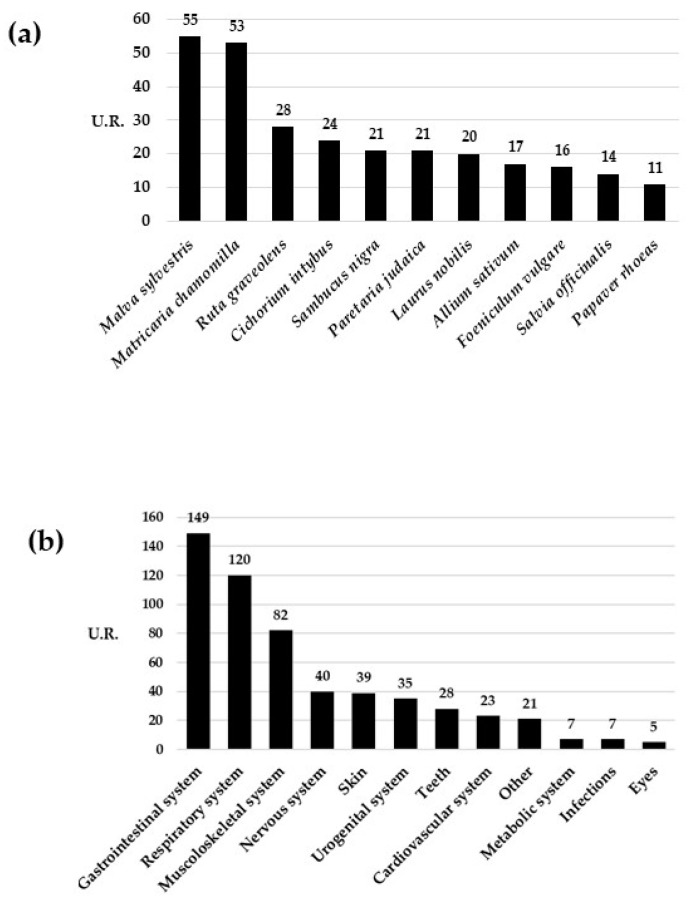
Number of use reports for each of the most cited species (**a**) and number of use reports for each ailment’s category (**b**).

**Figure 4 plants-12-03690-f004:**
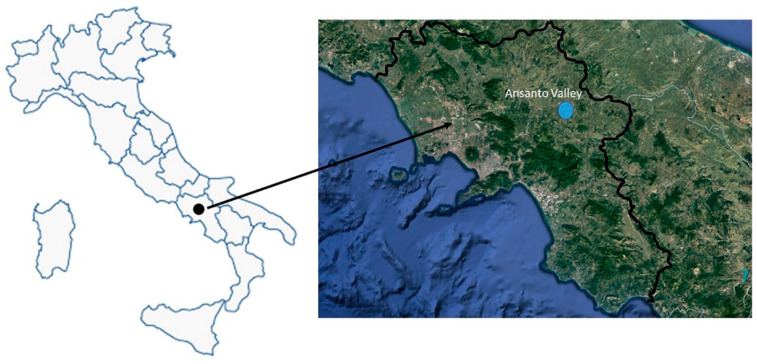
Geographical position of the Ansanto Valley (AV, Italy) (source: Google Earth, earth.google.com/web/ (accessed on 25 July 2023)).

**Figure 5 plants-12-03690-f005:**
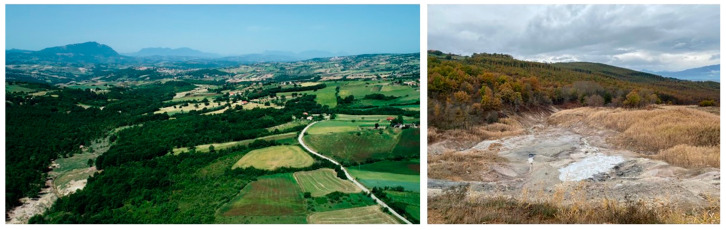
From left to right: Aerial view of the Ansanto Valley. Detail of the Mefite d’Ansanto area (Photos courtesy of Luigi Zollo).

## Data Availability

All the relevant data used for the paper can be found in Appendix A.

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
