# Peer review of "Ethnobotanical Documentation of the Uses of Wild and Cultivated Plants in the Ansanto Valley (Avellino Province, Southern Italy)"

_plants, 2023, doi:10.3390/plants12213690_

Round 1

Reviewer 1 Report

Comments and Suggestions for Authors

Well written contribution. 

Author Response

We thank the Reviewer for the positive comment.

Reviewer 2 Report

Comments and Suggestions for Authors

This is an excellent paper to be included in the indicated volume.

My only recommendation is to reorganize the paper into usual order of progression by putting the Methods before the Results.   This usual "order" provides the necessary context for understanding the results in the context of the environment in which the study was done.

Author Response

We thank the Reviewer for the positive reply.

The M&M chapter is before the References for Plants journal editorial rules

Reviewer 3 Report

Comments and Suggestions for Authors

Dear Authors,

My main concern here is the Introduction. What is the justification for the study on documentation of plant uses? Why are you doing this? What is the problem, etc. These points are missing in the Introduction.

I provided in the attachment some potential lines of argument!

Also revisit the conclusion following my comments in the attachment.

Comments on the Quality of English Language

English is ok with minor edits necessary.

Author Response

Please attached find our response to the Reviewer

Reviewer 4 Report

Comments and Suggestions for Authors

The article „Ethnobotanical research in the Ansanto Valley (Avellino province, southern Italy)“  analyzes the current knowledge about the usage of different Plant species in the mentioned region. I would like to thank the authors for their valuable contribution to the scientific discourse. In the following comments I would like to share my recommendations in the hope that they may give them valuable insights and improve overall quality.

At first, I will share some overall layout impressions, followed by major comments and ending with some minor comments.

The paper structure is well prepared, and the overall sections are in line with the journal’s technical recommendations. The research question of systematizing ethnobotanical knowledge is highly topical. The methodological approach of the snowball survey is already known and recognized in the field of ethnobotanical questions.

Major comments

Abstract

Line 11-13, in the context of the rest of the paper I would either recommend deleting this section or to specify in the associated section, why this environmental factor is of interest for your study. Does this unique geology have a direct influence on the usage of the plants? Are there any species that are only found in this specific environment? It would be really interesting if there are some causations. Furthermore, it would be interesting and enhance the point if there are any studies or self-performed experiments in which way the chemical profile of the plants changes under such conditions.

Line 19 the authors mention a total of 928 reports. On line 16 there is a total of 69 interviews mentioned. So please specify of which parameters the total numbers of 928 reports consists of. If there were any literature review work, please include this, with the methods used, in the abstract to.

Introduction

Line 62-64 the authors should change the term “Mephitis atmosphere” since it is not an official scientific term for the described phenomenon. It would be more suitable to write “the regions atmosphere” or a more appropriate term. Also as mentioned above it should be discussed why and in which way this circumstance has a direct influence on the authors work. This could be done with supporting chemical profiling data from literature of some of the plants, or through data from own experiments. It may also only briefly be mentioned in this section and then discussed in detail in the discussion section.

Results and Discussion

Line 79-80 the already mentioned 928 use reports should be urgent explained. There is no clear explanation of which these reports consist of. Are they only from literature? Are they from the Interviews or are they from both? It should be clearly stated of which these numbers consist of.

Line 85-87 Further discussion recommended. The data set seems to be interesting I would recommend and encourage the authors to dive deeper into it. How are the differences between literature and the interviews? Which confounders do the authors consider to be important? What limitations do the data have?

Line 95-96 this is again to simple. Great to see literature here to support the thesis. Are there any further explanations that may also explain the phenomenon? I would like to see more discussion on this point.

Line 99-102 more discussion needed. This is simple presentation of data points without extensive discussion. Why are primerly leaves used? Is there a link between the plant parts used and the administration?

Line 109-112 this set of indications maybe due to the age distribution of the used data sets. Since the average age of the key informants was 68 the mentioned uses could be due to typical health problems in this age group.

Line 118-210 the whole section lacks the scientific relevance the authors arose in the introduction. The simple naming of the mentioned plants with the mention of their use/indication (which are mostly already standard pharmaceutical knowledge) is not enough. In this section I would await extensive critical discussion of the findings from the interviews in contrast to the literature. A clear separation of the findings from the interviews (own data) to the literature data would be helpful and underline the further significance of the author’s work. Also, it would be of convenience to mention the exact number of the reports to each plant in the corresponding section. The reports are partially already mentioned in figure 3 but essential information is missing. Especially to the newly reported Sedum cepaea the number of the reports are of great interest. Corresponding to this number a classification would be of significance. Is a use report number of 57 better then one of 30? How is the quality in-between these reports? All in all, there is no clear mention of the influence of the relevance from the gaseous fumes from the area. This would be the perfect opportunity to underline the further significance of the work.

Line 211-256 the above-mentioned advice applies to this section too.

Line 257-272 section “Plants for ritual uses” this section seems to be out of scope. There is no clear interconnection between this section and the rest of the article. There is only one example use which is only partly of interest since there is no classification of the example ritual to the used plants in the area. If there are specific rituals corresponding to the used plants in the area it would be more suitable to mention these rituals with the discussion of the plant. Otherwise, I recommend deleting the section.

 Line 273-289 as already mentioned above the word “Mephitis” should be substituted with an actual geological scientific term. The section tries to highlight the importance of the high sulfur and CO2 atmosphere with the traditional use of the plants. There is no clear interconnection of this section with the plants used. A discussion of eventually different chemo profiles of the plants growing in the area with plants outside the area would be helpful to create this link.

Methods

Line 290-308 please state clearly why this area is of special interest. At this point the section arise the impression of a regional description.

Line 316-322 a clear naming of the authors inclusion criteria for the key informants would be of interest. Also, information about the structure of the snowball sampling would improve the overall quality. Are there any steps ore measures taken to verify that further informants meet the inclusion criteria? If yes, please state these measures.

Line 346-356 inclusion and exclusion criteria for the literature analysis should be stated.

Conclusion

Line 362-365 since there were no statement in the presented work on how the knowledge was passed down and even on the socio-economic development of the region, there is no basis for this resume.

Minor comments

Abstract

I would recommend spell checking with an English native speaker since there are some grammar and spelling mistakes

Introduction

Line 32-35 sentence has a logical break after the first comma, seems to be two different statements. I recommend finishing the first statement with a point instead of comma and start the second statement without direct interconnection to the first.

Line 36-38 should be more elaborated. Why is it such crucial part of ethnobotanical work to include the cultural aspect of the used plants? From a historic point it may be suitable to let this stand on its own. From a natural science point of view, it is often concluded from a cultural use to a current use. This means, for example, that either the culturally used plant parts are investigated or the applied extraction technique is included in the investigations. So this point should be elaborated more thoroughly in either way suitable.

Results and Discussion

Line 75-79 further discussion about the mentioned plant families needed. As the mentioned plant families form the largest European plant families, the increased occurrence in the use reports could be due to this alone.

Comments on the Quality of English Language

see above

Author Response

Please attached find our responses to the Reviewer comments

Round 2

Reviewer 3 Report

Comments and Suggestions for Authors

I recommend this be accepted for publication

Reviewer 4 Report

Comments and Suggestions for Authors

I thank the authors for their detailed response, numerous improvements, and explanations. In my opinion, the changes have significantly increased the quality of the article and contribute to the visibility of this extremely interesting and important field of research. I recommend publication of the article without further changes.